# Understanding Loss to Follow-Up in AMD Patients Receiving VEGF Inhibitor Therapy: Associated Factors and Underlying Reasons

**DOI:** 10.3390/diagnostics14040400

**Published:** 2024-02-12

**Authors:** Pavol Kusenda, Martin Caprnda, Zuzana Gabrielova, Natalia Kukova, Samuel Pavlovic, Jana Stefanickova

**Affiliations:** 1Department of Ophthalmology, University Hospital—St. Michael’s Hospital, 811 08 Bratislava, Slovakia; pavol.kusenda@gmail.com (P.K.); zuzkagabrielova@gmail.com (Z.G.); 2First Department of Internal Medicine, Faculty of Medicine, Comenius University, University Hospital Bratislava, 811 07 Bratislava, Slovakia; martin.caprnda@gmail.com; 3Vikom Ophthalmology Center, 010 08 Zilina, Slovakia; natalia.kukova9@gmail.com; 4Department of Ophthalmology, Faculty Hospital Nitra, 950 01 Nitra, Slovakia; samuelpavlovic@gmail.com; 5Department of Ophthalmology, Faculty of Medicine, Comenius University, 821 01 Bratislava, Slovakia; 6Oftalmocentrum Betliarska Euromedix, 851 07 Bratislava, Slovakia

**Keywords:** loss to follow-up, AMD, anti-VEGF, factors, reasons, non-persistence

## Abstract

Background: In patients with wet age-related macular degeneration (AMD), loss to follow-up (LTFU) leads to unplanned interruptions in therapy and the risk of visual loss. Methods: This retrospective and prospective case–control cohort study compared AMD patients with (LTFU YES) and without (LTFU NO) LTFU during anti-VEGF treatment over 12 years. LTFU was defined as missing any treatment or monitoring visits, or not scheduling follow-ups for six months. Results: Significant differences between LTFU NO (*n* = 298) and LTFU YES (*n* = 174) groups were age, treatment phase, baseline and final best-corrected visual acuity (BCVA), type of anti-VEGF drug, treatment switch, commuting distance, and escort during commuting. A multivariate logistic regression analysis identified the need for an escort during the commuting and treatment phase as the only significant difference. The four most common reasons for LTFU were general health worsening (21.8%), patient-missed appointments (16.7%), COVID-19-related issues (14.9%), and treatment dissatisfaction (8.6%). Conclusions: The factors associated with increased LTFU rates were older age, inactive treatment phase, lower baseline and final BCVA, bevacizumab treatment, monotherapy, longer travelling distance, and commuting with an escort. According to the multivariate logistic regression analysis, only the escort during the commuting and treatment phases was significant. These findings could direct research to explore social support in treatment adherence and highlight the importance of treatment phases in practice.

## 1. Introduction

Non-persistence refers to the discontinuation of therapy and is associated with visual loss in patients with wet age-related macular degeneration (AMD). We defined non-persistence (loss to follow-up—LTFU) in AMD patients treated with anti-vascular endothelial growth factor (anti-VEGF) therapy as missing any treatment or monitoring visits, or not scheduling follow-ups for six months.

The nAMD Barometer Leadership Coalition, an international group of experts in retinal disease, vision and diabetes care, and ageing, recently validated definitions for adherence and persistence to anti-VEGF therapy in AMD. Their definition for non-persistence is as follows: “not attending any treatment or monitoring visit for any reason within the last six months or not scheduling follow-up appointments for any reason for six months” [1]. The literature has various definitions of non-persistence and its synonyms: discontinuation, dropout, cessation, and loss to follow-up. 

In AMD patients treated with anti-VEGF, loss to follow-up increases the risk of vision loss [2]. The mechanism of vision loss is connected to new macular vessel formation that may leak and bleed. The vessels are accompanied by fibroblasts, disrupting the normal architecture of the retinal pigment epithelium (RPE)–photoreceptor complex, with a degenerative fibrovascular complex that ultimately produces a hypertrophic disciform scar [3]. 

Most antiangiogenic research in AMD has focused on the inhibition of vascular endothelial growth factor (VEGF), which increases in pigment epithelial cells in the early stages of AMD. VEGF is a homodimeric glycoprotein with a heparin-binding growth factor specificity for vascular endothelial cells. It induces vascular permeability, angiogenesis, and lymphangiogenesis, and it acts as a survival factor for endothelial cells by preventing apoptosis [3].

A systematic review and meta-analysis from Wong et al. reported disease progression outcomes for untreated patients with wet AMD, finding that the mean visual acuity change declined to 2.7 lines lost after 12 months and 4 lines lost after 24 months. The proportion of patients who developed visual acuity worse than 20/200 (practical blindness) increased from 19.7% at baseline to 75.7% by three years [4].

The unplanned discontinuation of treatment has been investigated in several studies, particularly during the COVID-19 pandemic [5,6]. The discontinuation of the treatment of wet AMD patients was found to have the worst impact on best-corrected visual acuity (BCVA)—they had the least chance among patients with retinal diseases treated with intravitre-al injections of BCVA returning to their original level after stopping treatment com-pared to diabetic macular edema (DME) and retinal venous occlusions (RVOs). Even patients with RVOs did not return to pre-LTFU levels regarding visual function [7].

Data from the IRIS registry in the United States indicate that one in seven patients with wet AMD are non-persistent with treatment. They noted non-persistence in 14.3% of patients, while older age, male sex, unilateral treatment, diabetes, Medicaid insurance, and Hispanic or Latino and Black or African American race or ethnicity were identified as risk factors [8].

According to a multicenter prospective cohort study from Germany, only a minority of patients (16.6%) with wet AMD were aware of the chronicity of their disease [9]. If patients are not aware that AMD is a long-term condition requiring ongoing treatment, they may be more likely to discontinue therapy prematurely, leading to non-persistence.

A recent systematic review gathered the available evidence on interventions that aimed to decrease LTFU in patients with AMD, RVO, and DME. Although some interventions have shown promise in glaucoma and diabetic retinopathy, no efficacious sole intervention has been developed to improve LTFU [10].

The broader aims and objectives of this study extend beyond the identification of factors associated with LTFU in patients with AMD treated with anti-VEGF. The study also seeks to understand the reasons for LTFU from the patients’ perspective. This dual approach not only provides a comprehensive understanding of the issue but also offers valuable insights into the patient experience.

The ultimate goal is to leverage these findings to develop strategies and interventions that can mitigate LTFU. By addressing the factors that contribute to LTFU and understanding the reasons behind it, healthcare providers can tailor their approaches to better suit the needs of the patients. This could lead to improved adherence to treatment, enhanced patient outcomes, and overall better management of AMD. Thus, the study serves a critical role in informing future research and practice in the field.

## 2. Materials and Methods

This is a retrospective (2010–2018) and prospective (2019–2022) case–control cohort study. The choice to conduct this type of study was motivated by the need to comprehend the patterns and factors associated with LTFU over a significant period of time. The data from the retrospective component were only obtainable retrospectively, as they were collected for other purposes and are being used in this study post hoc. On the other hand, the prospective component involves collecting data going forward, with a particular focus on observing data regarding LTFU. Since 2019, we have actively started capturing more specific LTFU data, which will enable us to gain a better understanding of the phenomenon and its underlying causes.

Our cohort reviewed patients who started anti-VEGF therapy for wet AMD at a single referral center over 12 years. Patients were treated and monitored at the Department of Ophthalmology, University Hospital—St. Michael’s Hospital, Bratislava, Slovakia. The availability of paper documentation in the application center was a condition for inclusion in the file.

Treatment regimens for anti-VEGF drugs were from the SPC of the drug used, which was used during the treatment period.

Ranibizumab therapy was started with an initial three doses of the drug at 4-week intervals. Up to 2015, it was followed by a reactive pro re nata (PRN) regimen based on the PrONTO study [11]. This included monthly follow-ups with the consideration of further administrations of the drug based on OCT and BCVA. For patients treated since 2015, we gradually switched to the proactive treat-and-extend (TREX) regimen, where an injection was given at each visit. This protocol was based on the TREX study [12]. After achieving the maximum and anatomical response according to OCT and BCVA, we extended the therapeutic intervals by two weeks; with the recurrence of activity, the interval was shortened by two weeks (maximum interval used was 12 weeks, minimum 4 weeks).

Aflibercept treatment was started with an initial three doses of the drug at 4-week intervals, followed by dosing every 8 weeks or the TREX regimen (intervals within 4–16 weeks). Extensions or shortenings were possible in 2- or 4-week intervals.

Brolucizumab treatment was started with three initial doses of the drug at 4-week intervals, followed by 8- or 12-week intervals according to disease activity.

Bevacizumab treatment was given in the same regimens as ranibizumab (three loading doses monthly, followed by PRN or TREX).

In the case where there were repeatedly no signs of disease activity at the maximum achieved interval (12–16 weeks), the treatment was suspended, and the patient was followed-up closely every 4–16 weeks. Treatment was reinitiated in the case of disease activity recurrence.

We recruited participants for our study from both paper and digital records of patients who had received intravitreal treatment at a healthcare center in the capital city of Slovakia. A total of 832 patients were assessed for eligibility based on the inclusion and exclusion criteria provided below, and out of them, 472 patients were enrolled for the study. The selection of patients for the study is illustrated in the flowchart in Figure 1.

Inclusion criteria:Initiation of anti-VEGF therapy (ranibizumab 0.5 mg, aflibercept 2 mg, brolucizumab 6 mg, or bevacizumab 1.25 mg) between 1 January 2010 and 1 January 2022;Treatment of wet AMD in accordance with the valid indication restrictions for this treatment;Administration of at least 1 intravitreal anti-VEGF injection;Minimum assessment period of 6 months since the first injection.

Exclusion criteria:Patients treated in interventional clinical trials;Patients who were administered other drugs (e.g., steroids) intravitreally;Intravitreal anti-VEGF treatment administered for reasons other than wet AMD (DME, RVO, proliferative diabetic retinopathy, visual impairment due to choroidal neovascularization of different etiology).

Detailed indications for anti-VEGF treatment initiation:Active subfoveal macular neovascularization (MNV) due to AMD;Baseline BCVA 20/25–20/200, in the case of a single seeing eye 20/25–20/320;Signs of MNV activity confirmed using optical coherence tomography (OCT) and/or fluorescein angiography (FAG) and/or optical coherence tomography angiography (OCT-A);In the case of active MNV with baseline BCVA worse than 20/200 or 20/320 in a single seeing eye, only off-label bevacizumab treatment was possible.

Loss to follow-up in AMD patients treated with anti-VEGF was defined as the presence of at least 1 interval without a visit to the application center (follow-up or intravitreal application) longer than 6 months. 

Patients were not considered LTFU if they had been scheduled to visit the application center for more than 6 months, had died within 6 months of the last visit, or had been referred to another site from the time of referral.

We collected data on the sex, the age of the patients at treatment initiation, patients’ residence and distance from the application center, time since the last injection, treatment phase, length of follow-up, initial and final BCVA, bilateral anti-VEGF treatment, type of anti-VEGF, treatment switch, and need for an escort when commuting to the center for treatment. In the context of patient care, an escort refers to a person who accompanies a patient during their commute to a healthcare center. This is often necessary when the patient is unable to travel alone due to physical, cognitive, or emotional challenges. The escort could be a relative, friend, or a professional caregiver. Their role is to ensure the patient’s safety and comfort during the journey. They may assist with tasks such as navigating the route, carrying personal items, or providing emotional support. In some cases, the escort may also facilitate communication between the patient and healthcare providers.

Patients who received an intravitreal injection within ≤16 weeks (≤112 days) from the last visit to the application center were considered in active treatment. If more than 16 weeks (>112 days) had passed since the last injection, the patient was in the phase of inactive treatment (follow-up). The approach is based on the maximum treatment intervals approved for the use of anti-VEGF drugs.

The duration of follow-up was determined by the period between the first intravitreal injection and the last visit to the application center.

Best-corrected central visual acuity was assessed using the Early Treatment of Diabetic Retinopathy Study (ETDRS) optotype in letter counts (as described in ETDRS study [13]). The vision of the treated eye was compared with that of the other eye. If the treated eye had a BCVA that was better by at least 6 letters, it was evaluated as the better-seeing eye. With a difference of up to 5 letters, it was a same-seeing eye. If the treated eye had a BCVA that was worse by 6 or more letters, it was the worse-seeing eye. If both eyes of the patient were treated with anti-VEGF, the parameter was not applied. 

We evaluated the change in BCVA as the difference in letters at the first anti-VEGF treatment and at the last visit. We created 5 categories of BCVA changes in the treated eye (according to the MARINA, ANCHOR, and VIEW trials [14,15,16]):(1)Deterioration (−6 to −14 letters)—this category represents a moderate decrease of visual acuity;(2)Improvement (+6 to +14 letters)—these patients have experienced a moderate gain in visual acuity, indicating a positive response to treatment;(3)Stabilization (+5 to −5 letters)—this category includes patients whose visual acuity has remained relatively stable, with minor gains or losses, where disease progression has been halted;(4)Super responders (+15 or more letters)—patients who have shown a significant improvement in BCVA, indicating a highly effective response to the anti-VEGF treatment;(5)Treatment failure (−15 or more letters)—this category represents a significant loss of BCVA, suggesting that the treatment has not been effective, or the disease has significantly progressed.

We also created 3 categories for the final BCVA of the treated eye at the last visit (in the case of bilateral treatment according to the better-seeing eye):(1)≥70 letters (≥20/40);(2)≥35 to ≤69 letters (20/200—20/50);(3)<35 letters (<20/200).

Patients were divided into two main groups—without loss to follow-up (LTFU NO) and with loss to follow-up (LTFU YES). In the LTFU YES group, we retrospectively searched the documentation for data on the reasons patients discontinued treatment and follow-up. From 2019, we either actively investigated these reasons at the return visit after LTFU, or we collected data using a telephone survey. Patients were asked “What is the main reason for the discontinuation of your treatment and follow-up?”

We compared patients below and above 75 years of age, as the median age of patients starting treatment was 75 years.

We also investigated the distance to the patient’s permanent residence from the application center. We divided patients into those who live in the same city as the application center (≤10 km) and those who live in another city (>10 km).

### Statistical Analysis

Data from analog and electronic medical records were captured and input into a Microsoft Excel data sheet and were then analyzed using IBM SPSS statistical software version 25. The normality of the distribution of the numeric data was checked using the Kolmogorov–Smirnov test. Since the distribution was not normal, the numeric data were further described by median and interquartile range. The difference between groups was analyzed using the Mann–Whitney U test. Parametric data were described by number and percentage, and the difference between groups was analyzed by Chi-squared test. We also conducted a multivariate logistic regression analysis.

## 3. Results

### 3.1. Characteristics of the Population

After considering the inclusion and exclusion criteria, 472 Caucasian patients aged 75.0 years [70.0–81.0] (186 men, 286 women) were included in the study.

### 3.2. Treatment

In the study, anti-VEGF drugs were mainly used as a monotherapy for treatment. The distribution of these drugs among the patients was as follows:51.9% (two-hundred and forty-five patients) were treated with ranibizumab;16.5% (seventy-eight patients) were treated with aflibercept;1.5% (seven patients) were treated with bevacizumab;1.7% (eight patients) were treated with brolucizumab.

In addition to this, there were some special cases:25.6% (121 patients) had their treatment switched to a different anti-VEGF drug during the course of the study.In 2.8% (13 patients), each eye was treated with a different type of anti-VEGF drug without switch.

The majority of patients (74.4% or 351 patients) received their treatment without any switch in the type of anti-VEGF drug.

The switch to another type of anti-VEGF drug was considered at the discretion of the attending physician in the case of suboptimal response to therapy. The most common reason for a switch was persistent or increased disease activity despite the shortest treatment interval.

There were 292 (61.9%) patients in the active phase of treatment and 180 (38.1%) in the inactive phase at the last recorded visit.

A total of 133 patients, representing 28.2% of the sample, received bilateral treatment, while treatment for one eye was needed for 339 patients (71.8% of the sample).

The median follow-up time was 47.0 [26.0–68.25] (LTFU YES) and 42.5 [16.0–72.25] (LTFU NO) months.

The median baseline BCVA was 60.0 [50.0–70.0] (LTFU NO) and 55.0 [37.8–68.3] (LTFU YES) letters. The median final BCVA was 64.0 [40.8–74.0] (LTFU NO) and 44.5 [26.0–72.0] (LTFU YES) letters.

### 3.3. Loss to Follow-Up

LTFU was recorded in 174 (36.9%) patients, while 298 (63.1%) patients were fully compliant during the monitored period. 

Reasons for LTFU were identified in 71.8% of patients. We were unable to find out the reasons for LTFU in the case of one patient’s death—his family did not respond to phone calls or address his refusal to communicate (not answering phone calls).

The reasons for LTFU were general health worsening (21.8%), patient-missed appointments (16.7%), COVID-19-related issues (14.9%), treatment dissatisfaction (8.6%), commuting problems (4.0%), reimbursement limitations (2.9%), family and personal matters (2.3%), and others (0.6%).

### 3.4. Commuting

Of the total, 248 patients (52.5%) resided in the same city as the application center, while the remaining 224 patients (47.5%) lived in different locations.

When it comes to commuting, 147 patients (31.1%) required an escort, while 193 patients (40.9%) were able to commute independently. For the remaining 132 patients (28%), it was not possible to determine whether an escort was necessary.

### 3.5. Factors

#### 3.5.1. Significant Factors

We identified several factors that are associated with LTFU. 

There is a significant difference in age distribution between the LTFU NO and LTFU YES groups. Out of all the patients under 75, a higher proportion (71.4%) were in the LTFU NO category than in the LTFU YES group (28.6%). On the other hand, of the patients over 75, a higher proportion (54.7%) were in the LTFU NO group than in the LTFU YES group (45.3%). Even without categorizing by age, the LTFU YES group is older on average, with a median age of 77.0 [71.0–83.0] years (Table 1 and Figure 2). 

The LTFU YES group also has a longer commuting distance, with a median of 22.0 [10.0–54.0] km, while the LTFU NO group lives closer to the application center, with a median of 10.0 [10.0–40.0] km. The effect of commuting distance is further evidenced by the fact that, of those living in the same city as the application center, a higher proportion were in the LTFU NO group than in the LTFU YES group. On the other hand, the LTFU YES group has a lower proportion of people living in the same city as the application center (30.6%) than those living elsewhere (43.7%) (Table 1 and Figure 3).

The need for an escort when commuting to receive treatment differs significantly between the LTFU NO and LTFU YES groups. Of those who did not need an escort, a higher proportion were in the LTFU NO group (83.9%) as compared to the LTFU YES group (16.1%) (Table 1).

The treatment phase also has a significant impact on whether an individual belongs to the LTFU NO or LTFU YES group. Of those in the active phase, a higher proportion were in the LTFU NO (72.9%) as compared to the LTFU YES group (27.1%) (Table 2 and Figure 4).

The time in days since the last anti-VEGF administration is longer in the LTFU YES group, with a median of 207.5 [0.0–1048.3], compared to 0.0 [0.0–199.5] in the LTFU NO group (Table 2).

The LTFU NO group also has a higher proportion of people who were treated with ranibizumab (57.1%), aflibercept (60.3%), or brolucizumab (87.5%) than those who received bevacizumab (28.6%). Conversely, the LTFU YES group has a higher proportion of patients treated with bevacizumab (71.4%) than those treated with ranibizumab (42.9%), aflibercept (39.7%), or brolucizumab (12.5%) (Table 2 and Figure 5).

There is a significant difference between the LTFU NO and LTFU YES groups regarding the switch of therapy from one type of anti-VEGF to another. The LTFU NO group has a higher proportion of switchers (75.2%) than non-switchers (59.0%). On the other hand, the LTFU YES group has a lower proportion of switchers (24.8%) than non-switchers (41.0%) (Table 2).

Baseline BCVA of the first treated eye was higher in patients in the LTFU NO group (60.0 [50.7–70.0]) than in patients in the LTFU YES group (55.0 [37.8–68.3]). The final BCVA of the first treated eye was also higher in patients in the LTFU NO group (64.0 [40.8–74.0]) than in patients in the LTFU YES group (44.5 [26.0–72.0]). Similarly, the change in BCVA of the first treated eye was greater in patients in the LTFU NO group (+1.0 [−13.0–10.3]) than in patients in the LTFU YES group (−2.0 [−20.0–7.0]) (Table 3).

The importance of the final BCVA and its change was also confirmed when considering the better-seeing eye in cases of bilateral treatment, where patients in the LTFU NO group had a higher median final BCVA of 69.0 [53.0–75.0] and a gain of +4.0 [−7.0–11.0], compared to patients in the LTFU YES group, who had a lower median final BCVA of 57.5 [30.0–73.0] and a gain of 0.0 [−13.3–8.3] (Table 3). 

The LTFU NO group has a higher proportion of people with a final BCVA ≥20/40 (71.2%) and ≥20/200 to ≤20/50 (64.5%) than those with <20/200 (42.9%). In contrast, the LTFU YES group has a lower proportion of people with a final BCVA ≥20/40 (28.8%) and ≥20/200 to ≤20/50 (35.5%) than those with <20/200 (57.1%) (Table 3 and Figure 6).

#### 3.5.2. Non-Significant Factors

The factors for which there were no significant differences between the LTFU YES and LTFU NO groups are shown in Table 1, Table 2 and Table 3.

#### 3.5.3. Multivariate Regression

We conducted a multivariate logistic regression analysis and identified the need for an escort during the commuting and treatment phases as the most influential factor. After adjusting for these two factors, the rest were not statistically significant (Table 4).

### 3.6. Underlying Reasons

#### 3.6.1. Overview

From the LTFU YES group (which makes up 36.9% of the entire group, *n* = 174), we were able to determine the reasons for LTFU in 71.8% of patients, in accordance with the study methodology.

#### 3.6.2. Common Reasons for LTFU

The most common known cause of LTFU was general health deterioration (21.8%) (Figure 7). 

The second most common reason was that the patient missed the appointment (16.7%). In these cases, the patient either forgot the exact date of the check-up or did not schedule an exact date during their last visit. 

The third most common known reason for LTFU (14.9%) was related to COVID-19, including restrictions on the movement of people (stay home policy), fear of contagion, or quarantine requirements.

#### 3.6.3. Treatment Dissatisfaction

Treatment dissatisfaction was also relatively common, with 8.6% of patients expressing discontent. Some patients cited various reasons for their dissatisfaction:Perceived insufficient effect of the treatment (1 patient);Fatigue from repeated and frequent visits (1 patient);Pain during application (1 patient);Long waiting times for application (2 patients);Loss of motivation for treatment (1 patient);No change in visual acuity (2 patients);The mental demands of the treatment (1 patient);Inability to see the purpose or benefit of the treatment (1 patient).

#### 3.6.4. Commuting Problems

Problems with commuting to treatment were cited as the reason for LTFU by 4.0% of patients, e.g., there was no one to bring them, they did not have the funds for the trip, the trip took too long, or they were unable to complete the trip using public transport on their own.

#### 3.6.5. Less Common Reasons for LTFU

Less common reasons for LTFU included issues with treatment reimbursement limitations imposed by the health insurance company (2.9%), family and personal matters (2.3%), and other reasons (0.6%). Among these other reasons, administrative problems related to treatment reimbursement requests were identified.

## 4. Discussion

### 4.1. Significant Findings

This study found several factors associated with LTFU in wet AMD patients undergoing anti-VEGF treatment. These include older age, inactive treatment phase, lower baseline and final BCVA, type of anti-VEGF drug, treatment switch, commuting distance, and the need for an escort during commuting. 

However, a multivariate logistic regression analysis identified the need for an escort during the commuting and treatment phase as the only significant factor. 

The four most common reasons for LTFU from patients’ perspectives were general health worsening, patient-missed appointments, COVID-19-related issues, and treatment dissatisfaction.

### 4.2. Age

Among the demographic factors, it is unsurprising that older patients are more likely to experience LTFU. As age increases, so does morbidity, often necessitating the assistance of others, such as during commutes for anti-VEGF treatment. 

Older studies did not consider higher age as an essential factor for non-persistence. Still, more recent data confirm our findings of an increased risk, especially in patients that are older than 80 and 90 [17,18,19,20].

### 4.3. Treatment Phase

The treatment phase was predictable, with patients in the inactive phase more often in the LTFU YES group. On the contrary, active treatment was protective for persistence. The explanation we suggest for this is because, in an inactive phase, the patient may feel that the treatment process is over and that further visits are unnecessary. This can be risky, especially in patients with an excellent response to anti-VEGF, when after repeatedly achieving the stability of the disease and the maximum intervals (12–16 weeks between injections, depending on the drug used [21,22,23]), we have to stop the treatment according to the current criteria for reimbursement from the public health insurance in the Slovak Republic, whereas it is possible to continue only after the reactivation of the disease [24]. This is also confirmed by the time since the last injection, which is significantly longer in the LTFU YES group compared to the LTFU NO group.

Some publications state that bilateral anti-VEGF treatment may be associated with a higher (odds ratio of 3.70 to 3.704) risk of non-persistence, but others also found it to be associated with a lower risk (odds ratio −0.69) [2,25,26]. This relationship was not statistically significant in our study.

Regarding bilateral anti-VEGF treatment, we found information in the literature about reducing the risk of non-persistence when injections were administered to both eyes on the same day [27]. We did not evaluate this parameter in our file, as our healthcare system does not officially allow this procedure. However, it could inspire a change in reimbursement, which could be protective for the persistence of our patients.

Among the treatment-related factors, we were surprised that the length of follow-up was not a decisive factor between the LTFU YES and LTFU NO groups. We expected that a longer follow-up period would be risky for patients due to fatigue, time loss, and repeated visits. A possible explanation is the positive perception of the functional visual benefit from the treatment, the professionalism of the medical team, the friendly atmosphere, and the organization of work, which many of our long-term treatment patients praise.

### 4.4. Best-Corrected Visual Acuity

From the point of view of visual acuity, our data on better baseline, final BCVA, and its improvement in the LTFU NO group compared to LTFU YES group point to the importance of the early detection of the disease and the immediate start of treatment to achieve better persistence and compliance. This is also shown in the resulting BCVA categories, where patients with excellent visual acuity, who are suitable for driving a motor vehicle in many countries, predominate in the LTFU NO group. Patients with lower but still relatively usable visual acuity are also highly represented in the LTFU NO group. On the other hand, patients whose BCVA results in practical blindness are at risk for non-persistence. From treatment success categories regarding BCVA change, we conclude that the patient evaluates the final visual acuity as more important for persistence than whether there was a change in BCVA. Even at first glance, less successful treatments (e.g., with high disease activity with a decrease in BCVA over time) can mean patient satisfaction, still acceptable visual functions, and a protective effect against LTFU when the injections are started early and the initial BCVA is higher.

The importance of a worse final BCVA as a factor negatively affecting persistence is documented in other works [25,28,29,30]. Moreover, some studies indicated an increased risk of non-persistence, even in patients without a BCVA change after anti-VEGF treatment [31,32,33].

### 4.5. Type of Anti-VEGF Drug

The surprising results are in the percentage distribution of used anti-VEGF in the LTFU YES/LTFU NO groups. The use of brolucizumab is shown to have a protective effect for reducing LTFU; on the contrary, treatment with bevacizumab is a risk factor. 

However, patients with a very advanced disease and low BCVA, who fall outside the indication criteria for on-label treatment, were mainly treated with bevacizumab.

Bias due to the low number of patients in monotherapy with these preparations cannot be ruled out either. It would be optimal to verify the high expectations for brolucizumab treatment in further research.

### 4.6. Treatment Switch

It was interesting that switching treatment in patients was a positive factor for persistence. Switching is most often performed when tachyphylaxis develops, in an attempt to achieve a more extended treatment interval or due to a suboptimal response to the original drug.

Since the beginning of the monitored period in Slovakia, on-label ranibizumab and off-label bevacizumab (not formulated for intravitreal use [34]) have been available for treatment. Since October 2013, treatment with aflibercept and, since August 2021, treatment with brolucizumab, have been covered. We mostly switched to preparations with less frequent dosing and a proactive treatment regimen. This may be the explanation for why the switch is protective against LTFU.

This also aligns with the knowledge that the proactive treat-and-extend regimen brings a lower treatment burden than the reactive PRN regimen [35].

### 4.7. Commuting Distance and the Need for an Escort during Commuting

Our data suggest that patients who live further away from the center or in another city are more likely to be non-persistent than patients who live closer. This is consistent with a study conducted by Obeid et al., who examined the distance from the application center—the odds of LTFU were lower for patients who lived ≤10 miles from a clinic [2]. We assume it could be connected with one of our study’s most important factors influencing persistence—the need for an escort during commuting. Patients who require an escort for commuting are at a significantly increased risk of being LTFU.

### 4.8. Reasons for LTFU from a Patient’s Perspective

The most common reason for non-persistence from a patient’s perspective (deterioration of general health status) is not surprising, given the average age of patients with wet AMD. 

The second most common reason was interesting for us—patients missed the appointment because they forgot the date or did not make an appointment in advance. In this case, a simple solution to the problem is offered—a reminder of the date (e.g., in the form of an SMS message, by phone, or in another electronic form) and an immediate assignment of the date of the next check-up/injection at each visit to the application center. It is not advisable to end the visit with a vague or tentative date (e.g., “come in 2 months”, “call later and make an appointment in 12 weeks”, or “wait for treatment approval”). From this point of view, the system of anti-VEGF reimbursement approval before treatment and its repeated continuation by the health insurance company is risky for non-persistence in Slovakia. 

The COVID-19 pandemic undeniably reduced persistence, as we noticed a higher frequency of LTFU during this period.

Compared to the same period before the COVID-19 pandemic, Chinese authors reported a 70% reduction in injections during the pandemic [36]. The COVID-19 pandemic caused delays in treatment with anti-VEGF injections for various conditions. BCVA is worse in patients without previous treatment and in those who need a continuation of treatment compared to patients without delays in treatment. This may have a long-term impact on the BCVA of patients requiring this vision-preserving treatment [37]. Other studies have also documented an unfavorable functional prognosis in patients after the discontinuation of anti-VEGF treatment in wet AMD and RVO [38,39].

Strict pandemic measures led to an increase in LTFU related to COVID-19. We also observed cases where there was no longer a medical reason for quarantine and postponing treatment. Still, despite this, patients were afraid to come for a long time (or relatives or caregivers from the nursing home did not let them go), primarily because of a media campaign that did not take into account the specifics of this long-term and regular biological treatment of chronic disease.

The IRIS registry does not capture disruptions in healthcare delivery and use that were caused by the COVID-19 pandemic, which likely worsened adherence [8]. In contrast, data from our study included the COVID-19 pandemic period.

Some of our patients expressed dissatisfaction with the treatment, as described in other works [31,40]. Possible solutions include new methods of therapy, forms of administration, or sufficient awareness about the disease.

Commuting problems as a primary reason for LTFU were rare. When considering the risks of an untreated disease and its progression to practical blindness, it is appropriate to consider the provision of transport for these people.

A minor reason for non-persistence was restrictions on treatment reimbursement by the health insurance company. These were primarily patients with high disease activity who required very frequent treatment. Currently, the public health insurance in Slovakia pays for a maximum of eight doses of anti-VEGF drug (ranibizumab, aflibercept, or brolucizumab)/eye during the first year of treatment and a maximum of six doses/eye in each subsequent year of treatment [24]. In the mentioned cases, the insurance company rejected further treatment in a given year after the maximum number of injections was exhausted. This led to a worsening of BCVA and demotivation of the patient.

The category of other reasons for LTFU was associated with an error in the request for treatment reimbursement as part of the high administrative burden of the medical staff.

A systematic review of risk factors for nonadherence and non-persistence to anti-VEGF treatment in wet AMD patients stated that non-persistence appears early, with almost 50% of patients discontinuing treatment within 24 months [41]. Similarly, nonadherence is frequently observed in 32% to 95% of patients. Definitions of nonadherence and non-persistence vary or were not given in many of the reviewed works. Several factors influence nonadherence and non-persistence—condition, therapy, patient, socioeconomic, and healthcare systems. Moderate-quality evidence points to lower baseline BCVA and poorer response to treatment as condition-related variables. The effect of other factors is less certain. Although many factors are uncontrollable (e.g., patient comorbidities), others are potentially solvable (e.g., lack of transportation or unrealistic patient expectations) [41].

Among Turkish wet AMD patients treated with ranibizumab on a PRN (pro re nata) regimen, 18.2% missed their monthly treatment visits, and 39.8% were unable to adhere to the optimal treatment regimen in the first year of therapy [31]. The patient’s compliance was influenced by better BCVA at the beginning of treatment, the smaller size of the lesion, residence near the hospital, higher education, sociocultural level, and financial status. The most frequent reasons for the discontinuation of treatment were fear of injection, lack of confidence in the benefit of therapy, financial limitations, continuation of treatment in another center, and systemic comorbidities [31].

In 2018, American authors evaluated LTFU in patients treated for wet AMD in a retrospective cohort study. They defined LTFU as the administration of at least one or more injections without a follow-up visit over 12 months. They found a high incidence of LTFU (22.2%) and identified multiple risk factors associated with LTFU. The probability of LTFU was greater among patients aged ≥81 years than those younger than ≤80 years. Regionally adjusted gross income also affected LTFU—lower-income patients were at a higher risk of LTFU. It was also important to know whether both eyes of the patient were treated with anti-VEGF treatment or only one eye—with binocular treatment, the chance of LTFU was lower [2].

We observed a relatively high LTFU rate, which we explain by the lengthy average follow-up period. 

In contrast to the IRIS registry [8], in the Slovak conditions of our study, the influence of male sex as a risk factor was not confirmed. Men and women were distributed equally in the LTFU YES and LTFU NO groups. The reason for this may be the high confidence of all patients, regardless of sex, in our therapeutic recommendations and free treatment reimbursement.

## 5. Study’s Limitations

The study is limited by its retrospective and prospective case–control design, which may introduce selection bias. Additionally, this study relies on patient self-reporting for reasons of loss to follow-up, which may be subject to recall bias or misreporting.

## 6. Brief Conclusions

The inactive treatment phase and the need for an escort during commuting are the most crucial factors in increasing the risk of LTFU in wet AMD patients undergoing anti-VEGF treatment. From the patient’s perspective, the primary reasons for LTFU mainly include a decline in general health, missed appointments, issues related to COVID-19, and dissatisfaction with the treatment.

## Figures and Tables

**Figure 1 diagnostics-14-00400-f001:**
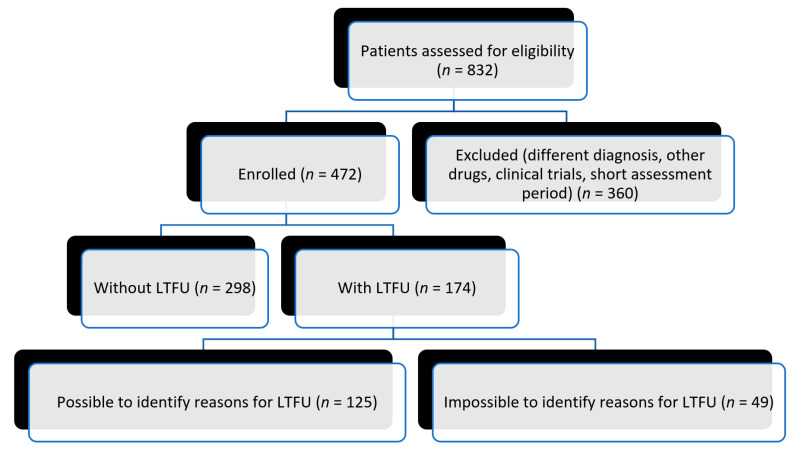
Flowchart—availability of patients in this study.

**Figure 2 diagnostics-14-00400-f002:**
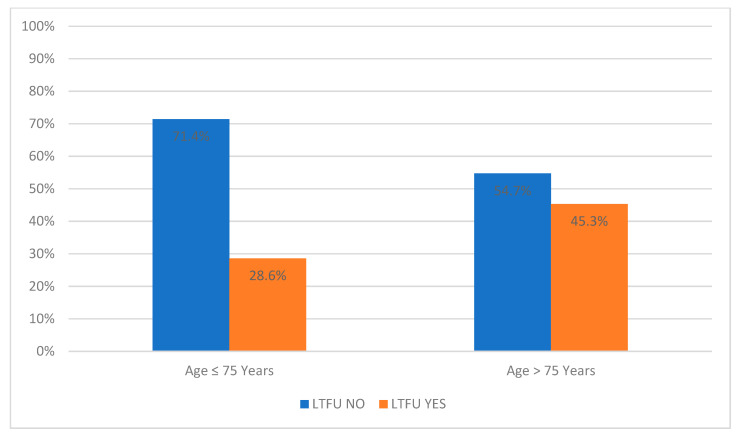
Percentage distribution of patients above and below 75 years in LTFU NO and LTFU YES groups. LTFU NO—patients without loss to follow-up, LTFU YES—patients with loss to follow-up.

**Figure 3 diagnostics-14-00400-f003:**
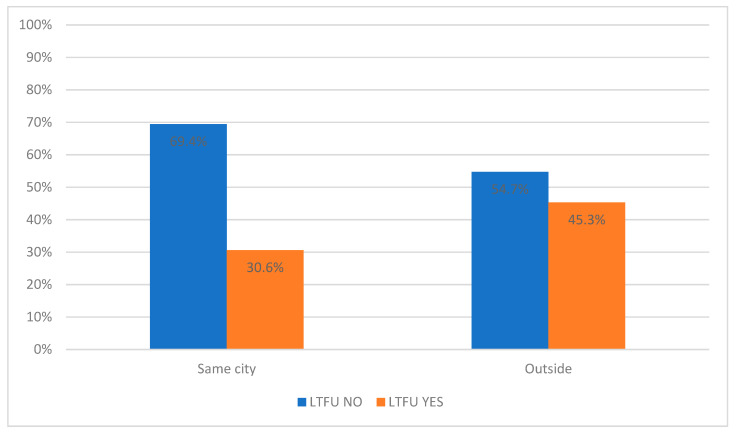
Percentage distribution of patients from the same city as the application center and outside it in LTFU NO and LTFU YES groups. LTFU NO—patients without loss to follow-up, LTFU YES—patients with loss to follow-up.

**Figure 4 diagnostics-14-00400-f004:**
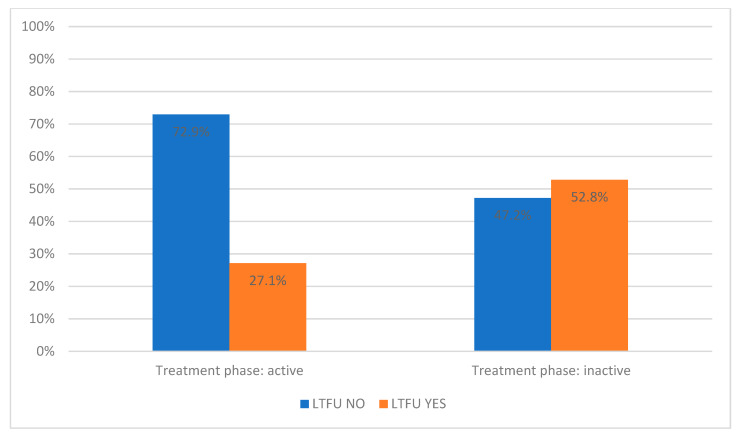
Percentage distribution of active and inactive treatment phases in LTFU NO and LTFU YES groups. LTFU NO—patients without loss to follow-up, LTFU YES—patients with loss to follow-up.

**Figure 5 diagnostics-14-00400-f005:**
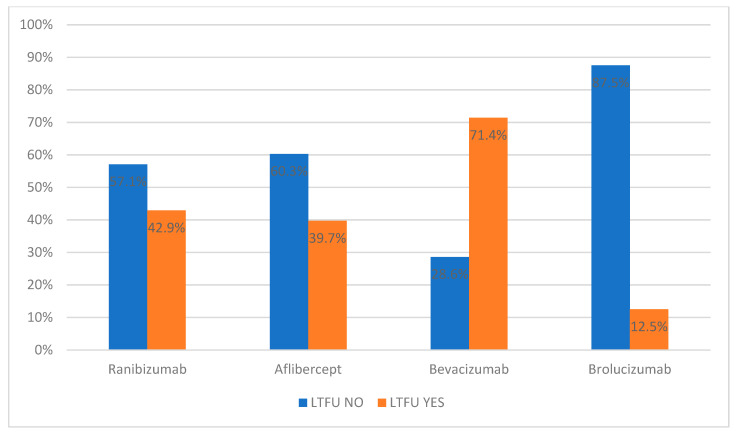
Percentage distribution of used anti-VEGF drugs in LTFU NO and LTFU YES groups. LTFU NO—patients without loss to follow-up, LTFU YES—patients with loss to follow-up.

**Figure 6 diagnostics-14-00400-f006:**
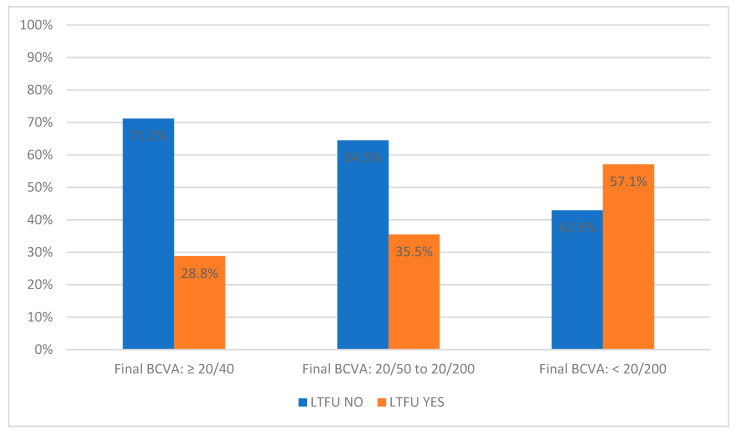
Percentage distribution of BCVA levels in LTFU NO and LTFU YES groups. LTFU NO—patients without loss to follow-up, LTFU YES—patients with loss to follow-up.

**Figure 7 diagnostics-14-00400-f007:**
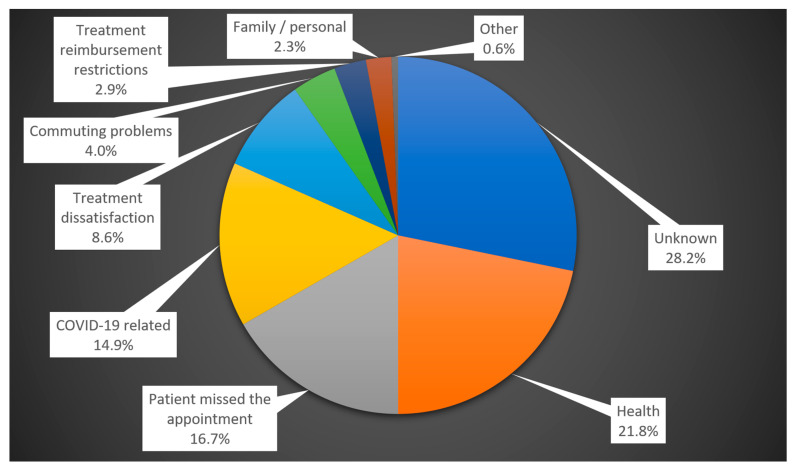
Reasons for LTFU.

**Table 1 diagnostics-14-00400-t001:** Demographic and commuting factors associated with LTFU.

Factor	LTFU NO	LTFU YES	Significance
Sex	Men: 118 (63.4% of all men)	Men: 68 (36.6% of all men)	*p* = 0.923 (NS)
	Women: 180 (62.9% of all women)	Women: 106 (37.1% of all women)	
Age	75.0 (69.0–80.0)	77.0 (71.0–83.0)	*p* = 0.003
Age above/under 75 years	≤75 years: 170 (71.4%)	≤75 years: 68 (28.6%)	*p* < 0.001
	>75 years: 128 (54.7%)	>75 years: 106 (45.3%)	
Patient’s residence in the same city as the application center/outside	Same city: 172 (69.4%)	Same city: 76 (30.6%)	*p* = 0.004
	Outside: 126 (56.3%)	Outside: 98 (43.7%)	
Commuting distance (km)	10.0 (10.0–40.0)	22.0 (10.0–54.0)	*p* = 0.004
Escort needed when commuting	Yes: 72 (49.0%)	Yes: 75 (51.0%)	*p* < 0.001
	No: 162 (83.9%)	No: 31 (16.1%)	
	Unknown: 64 (48.5%)	Unknown: 68 (51.5%)	

Values of *p* < 0.05 were considered significant. NS—not significant.

**Table 2 diagnostics-14-00400-t002:** Treatment-related factors and their association with LTFU.

Factor	LTFU NO	LTFU YES	Significance
Treatment phase	Active: 213 (72.9%)	Active: 79 (27.1%)	*p* < 0.001
	Inactive: 85 (47.2%)	Inactive: 95 (52.8%)	
Length of follow-up (months)	47.0 (26.0–68.3)	42.5 (16.0–72.3)	*p* = 0.228 (NS)
Time since last injection (days)	0.0 (0.0–199.5)	207.5 (0.0–1048.3)	*p* < 0.001
Anti-VEGF drug (non-switched patients) *	Ranibizumab: 140 (57.1%)	Ranibizumab: 105 (42.9%)	*p* = 0.001
	Aflibercept: 47 (60.3%)	Aflibercept: 31 (39.7%)	
	Bevacizumab: 2 (28.6%)	Bevacizumab: 5 (71.4%)	
	Brolucizumab: 7 (87.5%)	Brolucizumab: 1 (12.5%)	
Switch of therapy	Yes: 91 (75.2%)	Yes: 30 (24.8%)	*p* = 0.001
	No: 207 (59.0%)	No: 144 (41.0%)	
Bilateral anti-VEGF	Yes: 91 (68.4%)	Yes: 42 (31.6%)	*p* = 0.140 (NS)
	No: 207 (61.1%)	No: 132 (38.9%)	

Values of *p* < 0.05 were considered significant. NS—not significant. * This does not include the 13 patients who did not switch treatments and had both eyes treated with different anti-VEGF drugs.

**Table 3 diagnostics-14-00400-t003:** BCVA-related calculations and their association with LTFU.

Factor	LTFU NO	LTFU YES	Significance
Baseline BCVA (first treated eye)	60.0 (50.0–70.0)	55.0 (37.8–68.3)	*p* = 0.004
Final BCVA (first treated eye)	64.0 (40.8–74.0)	44.5 (26.0–72.0)	*p* < 0.001
BCVA change (first treated eye)	+1.0 (−13.0–10.3)	−2.0 (−20.0–7.0)	*p* = 0.017
Baseline BCVA (second treated eye in bilateral treatment)	70.0 (58.0–76.5)	66.5 (57.0–73.3)	*p* = 0.097 (NS)
Final BCVA (second treated eye in bilateral treatment)	70.0 (55.0–77.0)	66.5 (49.0–72.0)	*p* = 0.078 (NS)
BCVA change (second treated eye in bilateral treatment)	+0.0 (−7.3–7.0)	−1.0 (−13.0–5.0)	*p* = 0.531 (NS)
Final BCVA (in bilateral treatment, the better-seeing eye)	69.0 (53.0–75.0)	57.5 (30.0–73.0)	*p* < 0.001
BCVA change (in bilateral treatment, the better-seeing eye)	+4.0 (−7.0–11.0)	+0.0 (−13.3–8.3)	*p* = 0.015
Final BCVA—categories (in bilateral treatment, the better-seeing eye)	≥20/40: 141 (71.2%)	≥20/40: 57 (28.8%)	*p* < 0.001
	≥20/200 to ≤20/50: 118 (64.5%)	≥20/200 to ≤20/50: 65 (35.5%)	
	<20/200: 39 (42.9%)	<20/200: 52 (57.1%)	
Final BCVA of the treated eye versus the other	Worse: 126 (59.2%)	Worse: 87 (40.8%)	*p* = 0.316 (NS)
	Same: 29 (61.7%)	Same: 18 (38.3%)	
	Better: 58 (65.2%)	Better: 31 (34.8%)	
BCVA change—categories (first treated eye)	Treatment failure: 67 (54.5%)	Treatment failure: 56 (45.5%)	*p* = 0.137 (NS)
	Deterioration: 30 (65.2%)	Deterioration: 16 (34.8%)	
	Stabilization: 84 (62.7%)	Stabilization: 50 (37.3%)	
	Improvement: 65 (67.7%)	Improvement: 31 (32.3%)	
	Super responders: 52 (71.2%)	Super responders: 21 (28.8%)	

Values of *p* < 0.05 were considered significant. NS—not significant.

**Table 4 diagnostics-14-00400-t004:** Multivariate logistic regression analysis.

Parameter	B +/− S.E.	Wald Score	Exp(B) (95% CI)	Significance
Escort needed when commuting NO	-		1	*p* < 0.001
Escort needed when commuting YES	1.778 ± 0.266	44.704	5.915 (3.513–9.960)	*p* < 0.001
Escort needed when commuting UNKNOWN	1.596 ± 0.270	34.976	4.935 (2.907–8.375)	*p* < 0.001
Treatment phase ACTIVE	-		1	*p* < 0.001
Treatment phase INACTIVE	1.109 ± 0.217	26.130	3.030 (1.981–4.635)	*p* < 0.001

S.E. = standard error, CI = confidence interval. B = estimated coefficient, Exp(B) = exponentiation of the estimated coefficient.

## Data Availability

The data presented in this study are available on request from the corresponding author. The data are not publicly available due to privacy and ethical restrictions.

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
