# Peer review of "Understanding Loss to Follow-Up in AMD Patients Receiving VEGF Inhibitor Therapy: Associated Factors and Underlying Reasons"

_diagnostics, 2024, doi:10.3390/diagnostics14040400_

Round 1

Reviewer 1 Report (Previous Reviewer 1)

Comments and Suggestions for Authors

I have carefully reviewed the revised manuscript, and I am pleased to acknowledge the authors' diligent efforts in addressing the comments and making substantial improvements. The revisions have strengthened the overall quality of the manuscript, aligning it with the standards expected for publication. Consequently, I recommend accepting the manuscript for publication.

Author Response

Dear reviewer, 
We would like to express our sincere gratitude for taking the time to review our manuscript. Your positive and kind evaluation is greatly appreciated. We would also like to inform you that we have incorporated the suggestions and comments provided by the second reviewer and updated the manuscript accordingly. We hope you will appreciate the changes made, and we thank you again for your valuable feedback. 

Reviewer 2 Report (New Reviewer)

Comments and Suggestions for Authors

Although the topic is interesting, the structure of the manuscript need to be improved for each session (abstract, Method, Results, Discussion...)

Many points need to be clarify 

Review
Results section need to be structure : Characteristic of the population/ Treatment/ LTFU/ economic social category/ commuting

1.    « Reasons for LTFU were identified in 71.8% of patients”” clarity the reasons : financial burden,, transportation to the clinic, ineffective treatment feeling
2.    “Anti-VEGF drugs, such as ranibizumab (245 patients = 51.9%), aflibercept (78 patients = 16.5%), bevacizumab (7 patients = 1.5%), or brolucizumab (8 patients = 1.7%) were used for treatment. In 13 patients (2.8%), one eye was treated with one type of anti-VEGF and, simultaneously, the other eye with another type of anti-VEGF.: the sum does not achieve 100%. Please clarify
3.    Give de rate of treatment unsatisfying
Perceived insufficient effect of the treatment;
• Fatigue from repeated and frequent visits; •
 Pain during application;
• Long waiting times for application;
• Loss of motivation for treatment; •
No change in visual acuity; •
The mental demands of the treatment; •
 Inability to see the purpose or benefit of the treatment.

The discussion need to be reorganized : summarize the findings of the study first then discuss, point by point
Significant Findings need to be cited in the first paragraph of the discussion , then compare the results to other
 This study found several factors associated with LTFU in wet AMD patients undergoing anti-VEGF treatment. These include older age, inactive treatment phase, lower baseline and final best corrected visual acuity (BCVA), type of antiVEGF drug, treatment switch, commuting distance, and the need for an escort during commuting.

Author Response

Dear Reviewer,

We would like to express our sincere gratitude for taking the time to review our manuscript. We have carefully considered your comments and have made every effort to improve the quality of our work. Please find our responses to your comments and the revised version of the manuscript in the resubmitted files.
Thank you for your valuable feedback.

Round 2

Reviewer 2 Report (New Reviewer)

Comments and Suggestions for Authors

The manuscript is greatly improved and pleasant to read now

This manuscript is a resubmission of an earlier submission. The following is a list of the peer review reports and author responses from that submission.

Round 1

Reviewer 1 Report

Comments and Suggestions for Authors

Dear author,

Thank you for your manuscript titled " Understanding Loss to Follow-up in AMD Patients Receiving VEGF Inhibitor Therapy: Associated Factors and Underlying Reasons".

1. The manuscript text does not meet the standards of the English language. Several grammar and punctuation errors need correction.

2. Write the full term for each abbreviation upon its initial use. Ensure consistency in using abbreviations throughout the text (neovascular AMD, nAMD, wet AMD, BCVA, etc).

3. Please attempt to organize the abstract with the use of complete sentences.

4. The results of the study should not be mentioned in the "introduction" section (Lines 109-118).

5. Please specify the detailed indications for treatment initiation and switching treatments in the study (Lines 127 and 219).

6. It is better to specify the study's goal at the end of the "introduction" section, not the "results" section (Lines 202-204).

7. When discussing the final BCVA, it is important to mention the follow-up time and the number of injections or use adjustments for these variables (Line 213).

8. The sum of percentages in each group should add up to one hundred. Most of the statements in the results section need revision for better understanding of the results and manuscript clarity. For example, in Line 232: "In the group of patients under the age of 75, the LTFU-NO group had a higher proportion of patients than the LTFU-YES group (71.4% and 28.6%, respectively)." However, the original sentence may lead the reader to assume that in the LTFU-NO group, 71.4% of patients were under the age of 75.

9. It is better to mention the numeric results of some non-significant factors in the results section.

10. This study did not evaluate the anatomical outcomes of the patients (Line 431). There is potential selection bias in choosing different treatment regimens and anti-VEGF drugs if there was no consistent and clear treatment protocol for the physicians in the study. Concluding the efficacy of different treatment regimens and different anti-VEGF drugs is not appropriate given the study's design.

11. As the authors mentioned in the study, if the age of the patients is the most common reason for loss to follow-up, it is better to adjust for other risk factors related to the age of the patients.

12. At the end of the "discussion" section, it is better to provide the study's limitations, the most significant findings that contribute new information to the body of literature, and a brief conclusion.

appreciate the effort and time you invested in preparing the manuscriptI wish the authors success and suggest the above changes before submission to another journal.

Reviewer 2 Report

Comments and Suggestions for Authors

REview LTFU Diagnostics

1.     It is a very interesting subject and valuable material . However the text  needs proofreading by a professional language service. It is awkwardly written and does not “flow” smoothly.

2.     Please provide structured abstract with background, maethods, results, conlusions. Correct the syntax of the abstract ex. : LTFU patients in wet AMD not the other way round. (line 17)

3.     Please provide criteria of LTFU patients in the abstract.

4.     Line 21: significant factors – did authors mean significant differences ? Please be precise.

5.     Line 28: health ? Did authors mean problems with health ? Please be precise.

6.     Line 51 : provide the mechanism of vision loss in untreated wAMD patients.

7.     Line 54 : the least chance among patients with retinal diseases treated with intravitreal injections – be precise.

8.     Line 61: worse than … what ?

9.     Most of the information provided in the introductiuon: results of other studies on the subject, should be moved to discussion section. Also, authors provide some of their results at this point. Both of that should be corrected. Introduction should concentrate on mechanism of non-persistance in the treatment of wAMD. Authors should discuss other results in the context of their results in the discussion section.

10.  Line 122: please provide the aim of the study.

11.  Please provide the ethical board consent number, patient consent statemen and declaration of Helsinki statement.

12.  Line 130: 6 months after the first injection ? What about patients , who received ex . 3 loading injections and were followed altogether for just a six months ? We really do not have information about their compliance if the follow-up period is limited to 6 months only. Please think about study design. Maybe a year of observation time would be a better idea. (Especially in the context of the definition provided lines 144-146)

13.  Lines 131- 136 – these are not inclusion criteria but the detailed study design information about study protocol that should be provided elsewhere.

14.  I think that flow chart would better visualized the availability of patients in the study.

15.  Lines 192-200 – please create paragraph: Statistical analysis

16.  Line 208 – please provide the reason for not identifying the reasons for non-compliance.

17.  Line 209 – please provide range.

18.  210-211- active or not active at which point in time ?

19.  Line3 313- 326 – please do not repeat some of the data reported in tables.

20.  Please provide methodology on how were the data regarding reasons for LTFU collected (questionnaire ?). It has to be reported in the methods section.

21.  Discussion – please do not repeat results but comment on them. Move some of the considerations from the introduction here. Please refer to the regimens used for injections ( as I mentioned earlier, the regimens for anti-VEGF injections used in the study have to be described in detail)

22.  Lines 499-520 – I would remove it . Generally these are pure speculations. Please stick to your data. 

Comments on the Quality of English Language

English proofreading necessary.
